# Cytomegalovirus infection in malignant pleural mesothelioma

DeVon Hunter-Schlichting[1,2☯], Karl T. Kelsey[3], Ryan Demmer[1], Manish Patel[4], Raphael Bueno[5], Brock Christensen[6], Naomi Fujioka[4], Deepa Kolarseri[2], Heather H. Nelson[1,2☯]*

1 Division of Epidemiology and Community Health, University of Minnesota, Minneapolis, Minnesota, United States of America, 2 Masonic Cancer Center University of Minnesota Twin Cities, Minneapolis, Minnesota, United States of America, 3 Department of Epidemiology and Pathology and Laboratory Medicine, Brown University, Providence, Rhode Island, United States of America, 4 Division of Hematology and Oncology and Transplantation, University of Minnesota, Minneapolis, Minnesota, United States of America, 5 Division of Thoracic Surgery, Lung Center and International Mesothelioma Program, Brigham and Women's Hospital and Harvard Medical School, Boston, Massachusetts, United States of America, 6 Department of Epidemiology, Geisel School of Medicine at Dartmouth, Lebanon, New Hampshire, United States of America

☯ These authors contributed equally to this work.
* hhnelson@umn.edu

**Data Availability Statement:** All relevant data are within the manuscript and its Supporting Information files.

**Funding:** This work was supported by the National Institute of Health through grants awarded to NF

## Abstract

Human cytomegalovirus (HCMV) is a highly prevalent herpes virus which persists as a latent infection and has been detected in several different tumor types. HCMV disease is rare but may occur in high-risk settings, often manifesting as a pulmonary infection. To date HCMV has not been investigated in malignant pleural mesothelioma (MPM). In a consecutive case series of 144 MPM patients we evaluated two biomarkers of HCMV: IgG serostatus (defined as positive and negative) and DNAemia (>100 copies/mL of cell free HCMV DNA in serum). Approximately half of the MPM patient population was HCMV IgG seropositive (51%). HCMV DNAemia was highly prevalent (79%) in MPM and independent of IgG serostatus. DNAemia levels consistent with high level current infection (>1000 copies/mL serum) were present in 41% of patients. Neither IgG serostatus nor DNAemia were associated with patient survival. In tissues, we observed that HCMV DNA was present in 48% of tumors (n = 40) and only 29% of normal pleural tissue obtained from individuals without malignancy (n = 21). Our results suggest nearly half of MPM patients have a high level current HCMV infection at the time of treatment and that pleural tissue may be a reservoir for latent HCMV infection. These findings warrant further investigation to determine the full spectrum of pulmonary infections in MPM patients, and whether treatment for high level current HCMV infection may improve patient outcomes.

## Introduction

Malignant Pleural Mesothelioma (MPM) is a rare cancer that develops in the protective lining around the lungs and is tightly linked to asbestos exposure [1]. Once diagnosed, the prognosis is very poor, with a 5-year survival rate of 9% [1, 2]. Factors that contribute to worse patient

(R35 CA197292 and P30 CA77598) and KK (R01 CA126939). The Masonic Cancer Center also supported the study through a grant given to HN (R35 00052643). The funders had no role in the study design, data collection and analysis, decision to publish, or preparation of the manuscript.

**Competing interests:** The authors have declared no competing interests exist.

**Abbreviations:** HCMV, Human Cytomegalovirus; MPM, Malignant Pleural Mesothelioma; qdPCR, Quantitative Digital Polymerase Chain Reaction.

survival include advanced age, male gender, history of asbestos exposure, and non-epithelial histology subtype [3, 4]. Many patients with malignant mesothelioma ultimately die from pneumonia, respiratory failure or heart complications that are worsened by having mesothelioma [2, 5].

It has been suggested that the SV40 virus may contribute to MPM etiology, although the weight of evidence indicates it is not causal but may act as a co-factor [6]. This raises the question whether additional viruses might contribute to MPM. Human cytomegalovirus (HCMV) is a common pathogen with onco-modulatory properties [7–9]. HCMV is a member of the herpesvirus family and is a highly prevalent infection with approximately 50% of the population testing seropositive by age 50 [10]. HCMV infection is typically asymptomatic. However, in situations of immune suppression [11–13], including organ or stem cell transplant [14–16] and HIV [17, 18], infection can advance to HCMV disease, often manifesting as HCMV pneumonia [19]. Active HCMV infection has been described in approximately 30% of critically ill patients without a history of HIV or transplant [20]. In addition, HCMV DNAemia, indicative of an active viral infection, has been reported in patients with solid tumor malignancy [21–24].

The role of HCMV in cancer has primarily focused on the presence of virus in tumors [21, 22, 25–27]. Less well described is the epidemiology of active HCMV infection in solid tumor cancer patients. Whether HCMV, either at the tumor site or as an active DNAemia infection, is present in MPM and contributing to patient outcomes is an as-yet unexplored area of research. In this study we evaluated the prevalence of HCMV IgG seropositivity and HCMV serum DNAemia in MPM patients, as well as the prevalence of HCMV in tumor and normal pleural tissue. In addition, we evaluated whether these measures of HCMV were associated with patient survival.

## Materials and methods

### Study population

The study population has been previously described [28]. Briefly, study staff were embedded in the International Mesothelioma Program at the Brigham and Women's Hospital between May 2000 and May 2005. Staff approached patients at their initial clinic visit to participate in the research study. Eligibility requirements included age greater than 18 years, diagnosis of MPM based on pathology and ability to complete a questionnaire. At the time of consent into the study, trained interviewers worked with participants to complete a detailed questionnaire covering lifestyle habits, occupational history with an emphasis on possible encounters with asbestiform material, personal medical history, and other potential carcinogen exposures (questionnaire available upon request). Consent was obtained to abstract medical records, and a blood sample was obtained for use in research. The current study included 144 participants from the original epidemiologic study with sufficient available sample for CMV testing. This retrospective analysis using a deidentified dataset and specimens was reviewed by the University of Minnesota IRB and considered an exempt study (IRB: 0809E47928).

### Serum biomarkers of HCMV in MPM patients

HCMV IgG serology was performed on the COBAS e411 Analyzer (Roche Diagnostics, Indianapolis, IN) utilizing 150μL of serum, and an internal positive and negative CMV serological control. All IgG serology was performed at the Advanced Research and Diagnostic Laboratory (ARDL, Medical School, University of Minnesota, Twin-Cities). IgG status was categorized as negative or positive if the IgG value was over or below the threshold for reactivity (3 COI). To assess HCMV DNAemia, DNA was extracted from 200ul of serum using the Qiagen QIAamp DNA Mini Kit (Qiagen, Hilden, Germany). Serum samples were previously kept at -80°C until

DNA extraction. The standard DNA extraction protocol was amended to include a third column wash prior to the final elution of DNA into AE Buffer (Qiagen). 2μL of DNA were used in a dPCR reaction designed to quantify copies of the gBa). Primer sequences were `5'-TACC CCTATCGCGTGTGTTC-3'` and `3'-ATAGGAGGCGCCACGTATTC-5'`, and FAM/TAM probe (`5'-TTGCTGCCCAGCAGATAAGTGGTG`). These primers amplify a 254 bp product (82,475 to 82,728 bp). DNA was partitioned into a Fluidigm Biomark 37k IFC (Fluidigm Co. San Francisco, CA), which segregates 48 samples into 36,960 individual PCR reactions. PCR amplification, signal capture and quantification then occurred on the Biomark HD instrument and Digital PCR Analysis tool. Positive control DNA (Reference Material 2366a from the National Institute of Standards and Technology) and negative template controls were included on each IFC. All samples were run on duplicate arrays on different days, and discordant results run a third time. HCMV DNA concentrations were reported in copies/mL of serum. Prevalent HCMV was defined as DNA levels above 100 copies/mL which correlates with the assay's validated Limit of Detection (LoD). Patients with HCMV levels above 1000 copies/mL were categorized as having a high level current HCMV infection. Patients with levels below 1000 copies/mL were categorized as having a low level current HCMV infection.

## HCMV DNA in pleural tissue

DNA derived from frozen resected tumor was available for 40 MPM patients [29]. Normal pleural tissue samples, also frozen following resection, were obtained from the Brigham and Women's Hospital tissue bank from patients without mesothelioma, with no known asbestos exposure and no evidence of pleural disease (n = 21). These samples were from patients in the lung transplant program (both donors and recipients) or patients undergoing lung surgery for other disease processes that did not involve the pleura. In all control pleural samples, there was no gross or microscopic evidence of pleural pathology. DNA was extracted using the QIAamp DNA mini kit (Qiagen, Valencia, CA). Digital PCR for detection of HCMV DNA was as described above. The DNA extractions were not done with a standardized mass input, therefore results for tissue are reported as HCMV positive or HCMV negative.

## Lung cancer biobank samples

As a comparison group we evaluated serum from participants in the Masonic Cancer Center lung cancer biorepository; all with metastatic lung cancer. Beginning in March 2017 study staff monitored clinic schedules for patients that met eligibility criteria and approached them for participation in the biobank initiative. Eligibility requirements included adult patients with newly diagnosed, untreated metastatic lung cancer. A smoking history questionnaire was administered. Medical history, medication use, occupational and environmental exposure history, and demographic information were collected. All patients provided written, informed consent. A serum sample obtained at the time of participant consent (baseline visit) was tested for the presence of HCMV DNAemia (n = 22); all available samples in the biobank were tested. No a priori sample size calculations were performed, and all available samples were tested. The average age of the participants was 64 years (SD 9.8), 50% were women.

## Statistical analysis

Histology for MPM was assessed through pathology reports and classified as epithelioid or non-epithelioid (sarcomatoid or biphasic). The distribution of age (years), gender, history of asbestos exposure, and histology were compared across HCMV groups using t-tests and chi-square tests. Patient survival information was ascertained using the National Death Index and last known clinic visit. The relationship between HCMV and patient survival was evaluated

**Table 1. Clinical characteristics of MPM patients (n = 144).**

| Characteristic | HCMV DNAemia (copies/mL) | | | p-value |
|---|---|---|---|---|
| | <100 | 100–1000 | >1000 | |
| | Below LOQ | Low level infection | High level infection | |
| n (%) | 31 (21%) | 53 (37%) | 60 (42%) | |
| Age | | | | 0.67 |
| mean (SD) | 62.7 (±11.7) | 62.3 (±11.4) | 62.5 (±10.2) | |
| Sex | | | | 0.56 |
| Female | 7 (5%) | 9 (6%) | 9 (5%) | |
| Male | 22 (15%) | 44 (31%) | 51 (35%) | |
| Asbestos Exposure | | | | 0.22 |
| No | 6 (4%) | 9 (6%) | 17 (12%) | |
| Yes | 25 (17%) | 44 (31%) | 43 (30%) | |
| Histology | | | | 0.92 |
| Epithelioid | 22 (15%) | 36 (25%) | 39 (27%) | |
| Biphasic | 6 (5%) | 13 (9%) | 18 (13%) | |
| Sarcomatoid | 3 (2%) | 4 (3%) | 3 (2%) | |
| IgG | | | | 0.84 |
| Negative | 16 (11%) | 24 (17%) | 30 (20%) | |
| Positive | 15 (10%) | 29 (20%) | 31 (22%) | |

with the log-rank test comparing Kaplan-Meier survival probability plot strata. In addition, Cox proportional hazards models were utilized to adjust for potential confounding variables, and likelihood ratio tests were used to examine statistical significance. All statistical analyses were performed in R 3.6.3 and data was visualized using *ggplot2* and *ggfortify*.

## Results

The clinical characteristics of MPM patients included in this study are summarized in Table 1. Among the 144 patients, 113 were HCMV DNAemia positive (79%), and 59 (41%) had DNAemia levels indicative of an high level current HCMV infection (>1000 copies/mL serum). HCMV IgG seroprevalence was 52%, and serostatus was not associated with the presence nor level of HCMV DNAemia. In addition, age, sex, histology, and asbestos exposure were similar across HCMV DNAemia groups. The prevalence of HCMV DNAemia was similar in a comparison group of 22 metastatic lung cancer patients (80%), however, none of the metastatic lung cancer patients had HCMV viral loads greater than 1000 copies/mL.

Next, we evaluated the presence of HCMV DNA in both mesothelioma tumor and normal pleura tissues. DNA from tumor specimens collected at surgical resection was available for a subset of patients; 19/40 (48%) were HCMV DNA positive (Table 2). There was no association between tumor HCMV status and serum HCMV DNAemia, nor between tumor status and IgG seropositivity (Table 2). To evaluate whether HCMV infection was specific to malignant tissue, we assessed HCMV DNA in 21 normal pleura specimens from individuals without cancer; 6/21 (29%) were HCMV DNA positive (p = 0.47). Presence of CMV in the tumor was not associated with patient survival (S1 Fig).

Finally, we evaluated HCMV and patient survival. The median patient survival was 17.2 months. Median survival time was similar for those with no DNAemia (16.2 months) and those with high DNAemia >1000 copies/mL (15.6 months), while those with low level DNAemia had the longest survival time (21.1 months) (Fig 1). After adjusting for age, gender and histology there was no evidence for differences in survival for either low-level DNAemia (HR

**Table 2. Association between tumor and serum biomarkers of HCMV in MPM patients (n = 40).**

| Tumor HCMV DNA status | | | |
|---|---|---|---|
| | **Negative** | **Positive** | **p-value** |
| | **(n = 21)** | **(n = 19)** | |
| Serum DNAemia | | | *0.47* |
| < LOQ | 6 (60%) | 4 (40%) | |
| Low level infection | 10 (56%) | 8 (44%) | |
| High level infection | 5 (42%) | 7 (58%) | |
| IgG | | | *0.13* |
| Negative | 16 (48%) | 17 (52%) | |
| Positive | 5 (71%) | 2 (29%) | |

1.01, 95% CI 0.59–1.71) nor high-level DNAemia (HR 1.26, 95% CI 0.77–2.08) compared to those with no DNAemia. Similarly, IgG status was not associated with patient survival (HR 1.03, 95% CI 0.65–1.44) (S2 Fig).

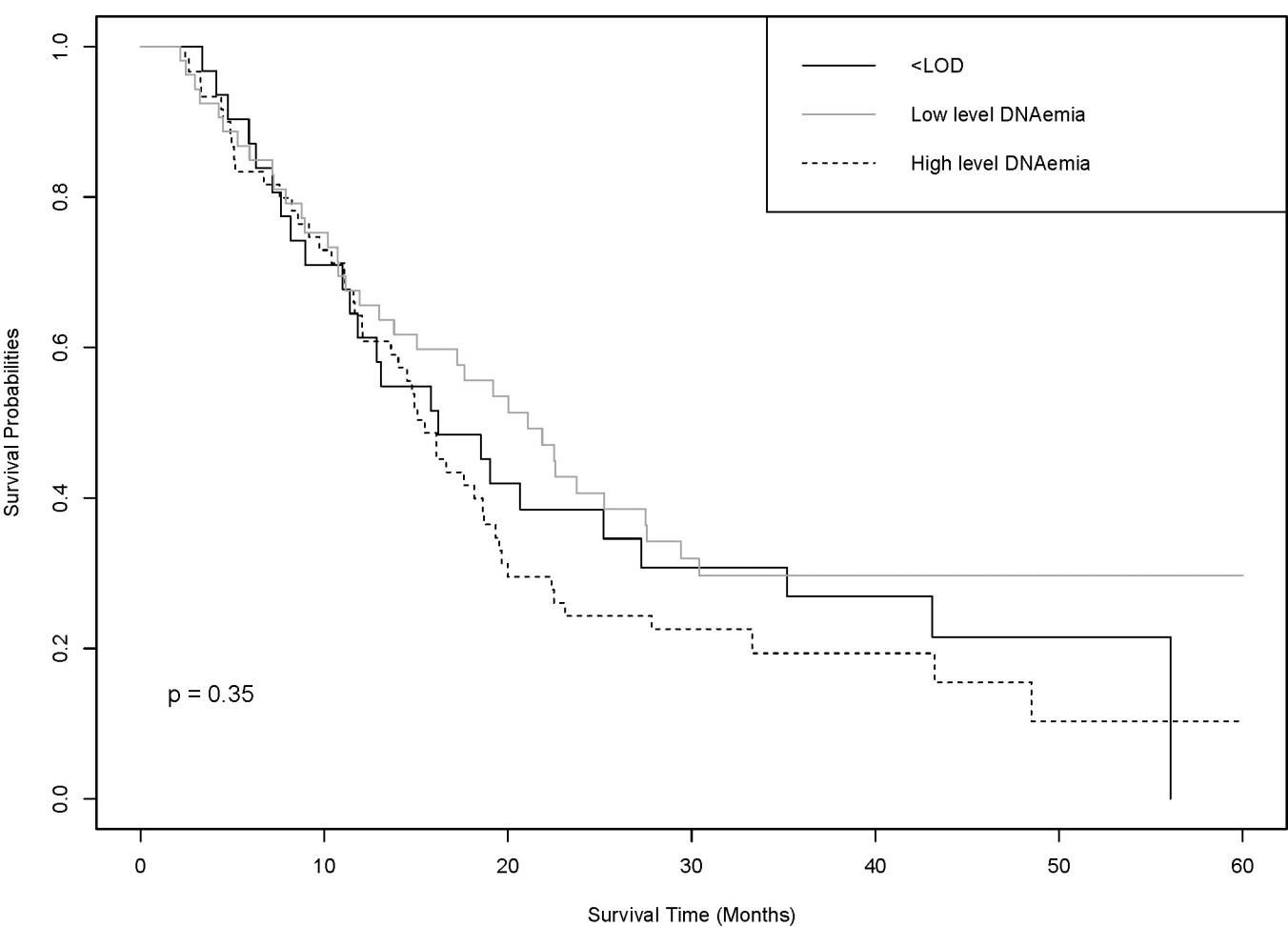

**Fig 1. HCMV DNAemia level and mesothelioma patient survival.** HCMV DNAemia levels >1000 copies/mL were associated with lower survival (15.6 months). A low level of HCMV DNAemia was associated with the longest survival (21.1 months). This association was not found to be statistically significant (p = 0.35).

## Discussion

We evaluated HCMV among patients with MPM and observed HCMV DNAemia viral loads indicative of an active HCMV infection (>1000 copies/mL) in 41% of patients. However, we did not observe evidence that this HCMV viremia was driven by infection at the tumor site. Our data suggest that active HCMV infection may be an unrecognized complication in MPM patients, putting them at risk for HCMV pneumonia as is observed in bone marrow and solid organ transplant recipients [14, 30–33].

HCMV, as assessed by IgG serostatus, is highly prevalent in the global population [34]. In healthy persons, HCMV infection is often asymptomatic. However, HCMV reactivation is of high concern in immune suppressed populations where it can progress to end-organ disease [20, 35], often presenting as a pulmonary HCMV infection (pneumonia) [19, 33], an observation confirmed in a MCMV mouse model [11]. Immune compromised populations, such as those with HIV, are also at risk for HCMV disease [18, 31, 36]. We suspect that inflammation, a known trigger for HCMV reactivation from latency, in combination with immune compromise, is contributing to HCMV viral activity in a subset of MPM patients. An important future direction of this research is to determine whether MPM patients are experiencing HCMV viral complications which could be treated.

HCMV DNAemia is intermittently prevalent in healthy populations (6–24%) [37, 38] and may occur due to primary HCMV infection, a second infection with a new HCMV strain, or reactivation of HCMV from latency. The limited reports on HCMV DNAemia prevalence in solid tumor patients suggest that prevalence may vary depending on the timing of blood collection, ranging from 11–93% [23, 24]. A significant limitation of the current work is a lack of standardization of blood draw in relation to course of treatment, making our estimate of DNAemia prevalence unstable. A 2010 study reported that among solid tumor cancer patients with CMV DNAemia, lung cancer was the most common diagnosis (30/75), however data on DNAemia levels were not available [39]. In the current study we compared HCMV DNAemia in MPM and metastatic lung cancer patients and found DNAemia common in both patient groups, but that high DNAemia levels (>1000 copies/mL serum) were specific to MPM. A limitation of our data is the inability to distinguish between a new infection or reactivation from latency. In fact, there was a striking lack of concordance between IgG status and DNAemia, which would suggest many patients were CMV naive and experiencing a primary CMV infection. Alternatively, CMV status may be misclassified by either the DNAemia or serology measurements. Studies from blood bank donation have mixed findings on this issue. Larsson et al reported discordance between IgG and DNAemia measurements, and suggested that a substantial proportion of the population may harbor CMV without developing a CMV IgG response [40]. In contrast, Roback et al. [41] did not observe discordance between DNAemia and IgG and claim observations of discordance are likely due to technical artifact. A comprehensive evaluation of CMV in a prospective study design is warranted.

We have established that in the absence of malignancy HCMV DNA is present in pleural tissue, indicating this may be a natural reservoir for latent infection. A higher proportion of tumor tissue was DNA positive, and this is consistent with the growing body of literature demonstrating the presence of HCMV DNA or viral proteins in the tumor environment [27], as well as multiple tissues serving as reservoirs susceptible to reactivation during times of immune suppression [40, 42, 43]. Across cancer types HCMV is present at low levels in the tumor, and it is not present as a clonal infection. Here, we tested a small amount of tissue, and given the non-clonal nature of tumor HCMV infections our reported prevalence of 29% is likely an underestimate. We did not observe an association between tumor HCMV status and

serum HCMV status. This too could be attributed to misclassification of tumor status given the small proportion of tumor material tested.

HCMV infection may be a biomarker that identifies patients who are the most immune-compromised or have the greatest tumor-associated inflammation. In addition, HCMV may be associated with immune benefit; there is a growing body of literature describing the emergence of adaptive Natural Killer (NK) cells with enhanced cytotoxic activity following HCMV reactivation [44, 45], and it is plausible that those with low level reactivation may have this immune benefit. Consistent with this hypothesis, those with high DNAemia had a shorter median survival than those with low level DNAemia. However, none of the observations regarding patient survival were statistically significant and further study is warranted. Specifically, capturing immune phenotypes and inflammation would provide clarity.

HCMV has been implicated multiple times in enhancing the malignancy of cancer cells and tumor associated cells. HCMV infection has been shown to modulate multiple molecular pathways involved in signal transduction of cellular activation [46, 47]. These onco-modulatory effects of HCMV on cellular metabolism are like those mediated by small DNA tumor viruses such as simian virus 40 (SV40) [6] and human adenovirus [48, 49]. In tumor cells HCMV-encoded regulatory proteins interfere with a variety of cellular signal transduction pathways leading to accelerated cell proliferation, enhanced survival, angiogenesis, cell motility and adhesion, thus enhancing the malignant behavior of tumor cells For this reason, potential treatment strategies such as immunotherapies using viral vectors such as HCMV could be employed and are currently being studied. The confirmation of the presence of HCMV is interesting as it provides a unique perspective that can guide future immunotherapies. There have already been promising results using CMV as a viral vector for many cancer immunotherapies [50].

A significant limitation of the present study is lack of treatment information. Patients were approached for study participation at a consultation visit. However, many did not seek treatment at the consenting hospital and treatment information is unavailable. An unexpected finding was the lack of association between HCMV DNAemia and IgG. This could occur if a subset of patients is HCMV naïve and experiencing a primary infection. Future work should include both replication in a second population of patients, and inclusion of IgM which would reflect new infections. Additionally, it would be useful to have bronchial swabs, or bronchial washing fluid (BAL) from each patient to confirm an active HCMV pneumonic infection. Lastly, no patient symptoms or co-infections were assessed at the time of blood draw. Osawa et al. 2009 reported that bacterial pneumonia was often present as a coinfection with HCMV viremia in solid tumor patients [20]. These shortcomings, and evaluation for HCMV pneumonia and bacterial infections, should be addressed in future work.

## Conclusions

The findings from the present investigation add to a limited but increasing evidence base supporting the use of identifying HCMV infection or reactivation in cancer patient care. Our data support the hypothesis that active HCMV infection is a common unrecognized clinical event among mesothelioma patients that may be associated with poor patient survival.

## Supporting information

**S1 Fig. HCMV DNA status in tumor tissue and mesothelioma patient survival.** MPM patients with HCMV DNA negative tumor tissue had a shorter survival period then MPM patients with HCMV DNA positive tumor tissue.
(TIF)

**S2 Fig. HCMV IgG status and mesothelioma patient survival.** Positive HCMV IgG status, indicating the presence of HCMV antibodies, was associated with lower survival when compared to those with negative HCMV IgG status. This association was not found to be statistically significant (p = 0.35).
(TIF)

## Acknowledgments

We would like to acknowledge and thank the Advanced Research and Diagnostic Laboratory (ARDL) located at the University of Minnesota for completing all serology services for this project.

## Author Contributions

**Conceptualization:** DeVon Hunter-Schlichting, Karl T. Kelsey.

**Data curation:** DeVon Hunter-Schlichting, Karl T. Kelsey, Raphael Bueno, Brock Christensen, Deepa Kolarseri.

**Formal analysis:** DeVon Hunter-Schlichting, Naomi Fujioka, Heather H. Nelson.

**Funding acquisition:** Heather H. Nelson.

**Investigation:** DeVon Hunter-Schlichting, Manish Patel, Brock Christensen, Heather H. Nelson.

**Methodology:** DeVon Hunter-Schlichting, Ryan Demmer, Heather H. Nelson.

**Project administration:** Heather H. Nelson.

**Resources:** Karl T. Kelsey, Ryan Demmer, Manish Patel, Raphael Bueno, Brock Christensen, Naomi Fujioka, Deepa Kolarseri, Heather H. Nelson.

**Supervision:** Ryan Demmer, Manish Patel.

**Validation:** DeVon Hunter-Schlichting.

**Visualization:** DeVon Hunter-Schlichting, Deepa Kolarseri.

**Writing – original draft:** DeVon Hunter-Schlichting.

**Writing – review & editing:** DeVon Hunter-Schlichting, Karl T. Kelsey, Ryan Demmer, Manish Patel, Raphael Bueno, Brock Christensen, Naomi Fujioka, Deepa Kolarseri, Heather H. Nelson.

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
