## [Decision Letter · Decision Letter 0]

23 Dec 2020

PONE-D-20-34162

Cytomegalovirus infection in malignant pleural mesothelioma

PLOS ONE

Dear Dr. Hunter-Schlichting,

Thank you for submitting your manuscript to PLOS ONE. After careful consideration, we feel that it has merit but does not fully meet PLOS ONE’s publication criteria as it currently stands. Therefore, we invite you to submit a revised version of the manuscript that addresses the points raised during the review process.

Two expert reviewers have commented on your article, and their feedback is appended below.  Overall the reviewers felt that results were publication-worthy and of interest to the scientific community, but additional information is needed before it can be published.  Three major issues need to be addressed:  1. greater detail on experimental methods overall, 2. improved presentation and discussion of DNAemia results, and 3. expanded discussion with respect to HCMV biology.  Specifically, more details on the methods is required, as noted by reviewer 1, particularly clarification on the sample preparation and analysis of HCMV serostatus.  In addition, both reviewers ask for more information and clarification on the DNAemia status, both in the results (Fig 1) and discussion sections, particularly in comparison to the general population.   Finally, the reviewers include several comments about typos, awkward working, or areas where enhanced or clarified discussion of the results is necessary.  Your revised manuscript should effectively address each of the the reviewer's comments below.

We look forward to receiving your revised manuscript.

Kind regards,

Juliet V Spencer, Ph.D.

Academic Editor

PLOS ONE

Journal Requirements:

2. In your Methods section, please provide additional information about the participant recruitment method and the demographic details of your participants in both cohorts. Please ensure you have provided sufficient details to replicate the analyses such as: a) the recruitment date range (month and year), b) a description of how participants were recruited.

3. Please provide a sample size and power calculation in the Methods, or discuss the reasons for not performing one before study initiation.

4. Please include additional information regarding the survey or questionnaire used in the study and ensure that you have provided sufficient details that others could replicate the analyses. For instance, if you developed a questionnaire as part of this study and it is not under a copyright more restrictive than CC-BY, please include a copy, in both the original language and English, as Supporting Information.

5. Please note that PLOS does not permit references to “data not shown.” Authors should provide the relevant data within the manuscript, the Supporting Information files, or in a public repository. If the data are not a core part of the research study being presented, we ask that authors remove any references to these data.

6.Thank you for stating the following in the Acknowledgments Section of your manuscript:

"This work was supported in part by NIH P30 CA77598 utilizing the Masonic Cancer Center,

University of Minnesota shared resources and the sponsored by the R35 grant (00052643)"

7. Please include a caption for figure 1.

Reviewers' comments:

Reviewer's Responses to Questions

**Comments to the Author**

1. Is the manuscript technically sound, and do the data support the conclusions?

Reviewer #1: Yes

Reviewer #2: Partly

2. Has the statistical analysis been performed appropriately and rigorously? 

Reviewer #1: Yes

Reviewer #2: Yes

3. Have the authors made all data underlying the findings in their manuscript fully available?

Reviewer #1: Yes

Reviewer #2: Yes

4. Is the manuscript presented in an intelligible fashion and written in standard English?

Reviewer #1: Yes

Reviewer #2: Yes

5. Review Comments to the Author

Reviewer #1: Hunter-Schlichting and co-authors present an interesting preliminary study assessing the incident of HCMV infection in malignant pleural mesothelioma patients. While the study is small, the authors are reasonable in their assessment of the data and their conclusions, and fully outline the limitations to the patient cohort and reasonable analysis thereof. This is an interesting preliminary study, and the authors properly address all aspects of the patient cohort, analyses, and conclusions, with an emphasis on future study. This paper provides a useful addition to the literature and is scientifically sound. To that end, I have only a few minor comments:

1) The finding that more patients have positive DNAemia than are seropositive is surprising and quite strong even using a smaller study sample size. This finding could be highlighted more as this would be a very interesting follow up.

2) On the same note as point 1 above, the conclusions/discussion regarding a latent HCMV infection contributing to disease are not supported by the data presented here. The data suggests - both by higher DNAemia patients (although any detectable DNA typically suggests a non-latent viral state in patients) trending towards increased mortality, and the by the lack of detectable IgG in DNA+ patients; that an initial or reactivated viral infection is more likely associated. The authors do address this in the Discussion, but the emphasis on latent infection is overstated.

3) Why was the DNAemia reporting level changed between patient cohorts (page 7)? Serum was analyzed for DNA copies and grouped into three categories (undetected, DNA+, active infection (greater than 1000 copies/mL) - so negative, positive, active. Yet tissue samples, although run in the same manner were categorized as only negative or positive. Active infection could be defined using similar criteria here as in serum (i.e. greater than 1000 copies per ug tissue for example)? Or discuss limitations as to why not? This is relevant to the Discussion as well (third paragraph, last sentence), which states that "viremic levels (>100 copies/mL serum) were specific to MPM" which is not supported by the data in this cohort, although I agree that it does warrant further study in a larger cohort.

4) Additional information should be included as to whether patients were assessed for treatment status / immunosuppression at the time of sampling (table 1).

5) The prevalence of HCMV in these samples is a key piece of data. The data presented on page 8 (Table 1) however suggests that prevalence of HCMV is comparable to the general population and DNAemia is similar in similar sampled tissues (metastatic lung cancer, data not shown). This is a very key point to discuss since this could suggest that the findings of HCMV prevalence are no different than background population levels. This could be clarified in the Discussion where a couple of statements (i.e. third sentence of paragraph one - "suggest HCMV infection may be an unrecognized co-morbidity") are overstated.

6) There are some minor typos/formatting/missing punctuation, including in Table 2.

Reviewer #2: In the article entitled, “Cytomegalovirus infection in malignant pleural mesothelioma”, by Hunter-Schlichting, et al, the authors describe the prevalence of CMV in patients with malignant pleural mesothelioma (MPM). To the authors’ and this reviewer’s knowledge, this is the first investigation into the potential connection between CMV and MPM. Overall, the manuscript could be clarified in some places, which should not prove overly cumbersome but will undoubtedly improve readability. Also, some of the data should be more clearly presented (e.g. Fig 1), so the reader can appropriately interpret the data herein. Suggestions are provided below for the authors’ consideration:

Major:

1. Clarity to the methods is warranted.

a. Study population: how long between the time of consent and blood sampling?

b. Serum biomarkers: Stating “…serology was performed on the Roche e411 Analyzer…”, with no additional information as to how the samples were prepared, etc. is definitely not enough information. Similarly, how were the tissue samples prepared? In the same paragraph, what does “gBa region” refer to? Is it not just “gB”? Was an uninfected control used to determine baseline for the CMV DNA qPCR? This would allow one to determine if the primers detect background (in other words, it allows one to determine the baseline). This could also be why there was no association between IgG and DNAemia (which the authors describe as ‘unexpected’ in their Discussion, p13).

c. HCMV DNA in tissue: Were the normal tissue samples also freshly frozen? What were the methods used to extract the DNA from tissue?

d. Lung Cancer BioBank samples: “Blood was collected at specified time points during a patient’s treatment course” - Did all patients receive the same treatment course, and did treatment regimens differ between the two study sites? What were these specified time points?

2. Figure 1 is hard to evaluate. The lines are not labeled, making it difficult to interpret.

3. p11 – “HCMV…is highly prevalent in the global population and nearly ubiquitous in immunocompromised populations” – I would argue this is misleading, especially since the authors use the term ‘immunocompromised’ to describe transplant patients as well (who are technically immunosuppressed). Likewise, “HCMV infection may be a biomarker that identifies patients who are the most immune-compromised…” is misleading.

4. p12 – “…those with the longest survival had low level DNAemia” – Isn’t this not completely true when adjusted for age? Similarly, p14 (Conclusion section) – “…MPM patients with a negative or low level HCMV infection were greater than those with a high level HCM infection” – isn’t this not the case when adjusted for age? (Also, minor, but this last sentence should read “…HCMV infection”).

Minor:

1. For ease in reviewing, it would help a ton to have line numbers.

2. Various points in the manuscript should be clarified:

a. What is meant by “HCMV…has been observed as viremia in various cancer patients and the tumor environment in different cancers” (p4)?

b. p4 – “>50% of the population infected by ages 75-80.” This should be double-checked. It’s more like >50% by 40 years of age.

c. What is meant by “Less is known about the experience of HCMV viremia in cancer patients…” (p4-5)?

d. The data re: 21 normal pleura samples – this is not included in the tables is it?

e. p10, p11, p12 – “HCMV biomarkers” should actually be defined.

f. p12 – what is meant by a “robust NK cell phenotype”?

3. Figures should be presented in order of appearance. As is, the supplemental figures need to be reversed. Also, I would suggest the authors put the supplemental figures in the main body of the manuscript as opposed to including them as supplemental figures.

4. The authors should include an overall “take-home” message for their findings at the end of the results section. As is, this section just drops off, leaving the reader wondering what the overall take is.

5. p11 – The authors state that HCMV infection could put MPM patients at risk for HCMV-associated pneumonia. Do any of these patients the authors evaluated present with this symptom?

6. p12 – I appreciate the authors’ idea that pleural tissue may serve as a site of latency. But it may also be worth noting that infiltrating, latently infected monocytes that are recruited to the ‘injured’ pleura could be a source of reactivating virus in this tissue. Just a thought.

6. PLOS authors have the option to publish the peer review history of their article (what does this mean?). If published, this will include your full peer review and any attached files.

Reviewer #1: No

Reviewer #2: No

---

## [Author Response · Author response to Decision Letter 0]

14 Jun 2021

Seven issues were raised by the academic editor. A response to each of these issues is provided below.

1. Please ensure that your manuscript meets PLOS ONE’s style requirements, including those for file naming. 

This has been done.

2. In your Methods section, please provide additional information about the participant recruitment method and the demographic details of your participants in both cohorts. 

We have added additional text to the methods section describing participant recruitment for both the mesothelioma study and the lung cancer biobank. The demographic characteristics of the lung cancer biobank participants are now included in the text of the methods. 

Please ensure you have provided sufficient details to replicate the analyses such as: a) the recruitment date range (month and year) and b) a description of how participants were recruited. 

These details have been added to the methods section.

3. Please provide a sample size and power calculation in the Methods or discuss the reasons for not performing one before study initiation.

No a priori sample size calculations were performed. We tested all available samples from the mesothelioma study and lung cancer biobank. A statement to that effect was present in the original description of the mesothelioma study population, and a similar statement has now been added to the lung cancer biobank description.

4. Please include additional information regarding the survey or questionnaire used in the study and ensure that you have provided sufficient details that others could replicate the analyses. For instance, if you developed a questionnaire as part of this study and it is not under a copyright more restrictive than CC-BY, please include a copy, in both the original language and English, as Supporting Information.

The epidemiologic study was conducted over 20 years ago and the questionnaire is not available. The data used in this study are not specific to the questionnaire, with the exception of asbestos exposure. In our analysis we utilized self-report of asbestos exposure and it was not associated with CMV status. The other variables investigated (age, sex, and histology) are easily replicated in a clinical study using data from the health record.

5. Please note that PLOS does not permit references to “data not shown.” Authors should provide the relevant data within the manuscript, the Supporting Information files, or in a public repository. If the data are not a core part of the research study being presented, we ask that authors remove any references to these data.

The data are now more fully described in the text of the manuscript, and “data not shown” has been removed.

6. Thank you for stating the following in the Acknowledgments Section of your manuscript: "This work was supported in part by NIH P30 CA77598 utilizing the Masonic Cancer Center, University of Minnesota shared resources and the sponsored by the R35 grant (00052643)" We note that you have provided funding information that is not currently declared in your Funding Statement. However, funding information should not appear in the Acknowledgments section or other areas of your manuscript. We will only publish funding information present in the Funding Statement section of the online submission form. Please remove any funding-related text from the manuscript and let us know how you would like to update your Funding Statement. Currently, your Funding Statement reads as follows:"The funders had no role in study design, data collection and analysis, decision to publish, or preparation of the manuscript." Please include your amended statements within your cover letter; we will change the online submission form on your behalf.

We have removed from the funding information from the Acknowledgments Section of the manuscript and have an updated Funding Statement in the cover letter. It is repeated here: “This work was supported by the following NIH grants: R35 CA197292, R01 CA126939 and P30 CA77598. The funders had no role in the study design, data collection and analysis, decision to publish, or preparation of the manuscript".

7. Please include a caption for figure 1.

A caption has now been added to Figure 1.

Reviewer #1

1) The finding that more patients have positive DNAemia than are seropositive is surprising and quite strong even using a smaller study sample size. This finding could be highlighted more as this would be a very interesting follow up.

We agree that the lack of concordance between DNAemia and serostatus is surprising. Among our patient population it is possible that patients were CMV naive and are now experiencing a primary CMV infection. Alternatively, CMV status may be misclassified, either due to technical artifact or lack of antibody response. We have broadened our discussion of DNA-IgG discordance to describe these possibilities and present our finding in context of what has been observed in the literature. Specifically, there are two large observational studies from blood bank donation centers. One observed discordance similar to what we observe in our study, and these authors suggest that a substantial proportion of the population may be “CMV carriers” who have CMV but did not develop an IgG response. The second observed no discordance and suggested that if discordance is observed it is likely a technical artifact or arises due to new primary infection.

2) On the same note as point 1 above, the conclusions/discussion regarding a latent HCMV infection contributing to disease are not supported by the data presented here. The data suggests - both by higher DNAemia patients (although any detectable DNA typically suggests a non-latent viral state in patients) trending towards increased mortality, and the by the lack of detectable IgG in DNA+ patients; that an initial or reactivated viral infection is more likely associated. The authors do address this in the Discussion, but the emphasis on latent infection is overstated.

We agree with the reviewer that we need to be more careful with our language regarding CMV status throughout the manuscript and recognize that we cannot distinguish between primary and reactivated infection. Based on the reviewers comments we now refer to “low level current infection” and “high level current infection” with regard to the serum DNAemia data. The manuscript has been edited accordingly.

3) Why was the DNAemia reporting level changed between patient cohorts (page 7)? Serum was analyzed for DNA copies and grouped into three categories (undetected, DNA+, active infection (greater than 1000 copies/mL) - so negative, positive, active. Yet tissue samples, although run in the same manner were categorized as only negative or positive. Active infection could be defined using similar criteria here as in serum (i.e. greater than 1000 copies per ug tissue for example)? Or discuss limitations as to why not? This is relevant to the Discussion as well (third paragraph, last sentence), which states that "viremic levels (>100 copies/mL serum) were specific to MPM" which is not supported by the data in this cohort, although I agree that it does warrant further study in a larger cohort.

For the serum data it is possible to back-calculate the dPCR results as copies/mL serum. However, for the tumor data we were not confident in making a similar calculation, largely because the input tumor mass for the extractions was not constant across samples. There is a large clinical body of literature that equates serum/plasma DNAemia with the presence of a current replicating CMV infection. The same is not true with the tumor tissue. For these reasons we felt most comfortable classifying the tumor as positive or negative. We have now included this information in the methods section.

4) Additional information should be included as to whether patients were assessed for treatment status / immunosuppression at the time of sampling (table 1).

We acknowledge that this is an important issue. Unfortunately, treatment status information is not available on this study. We have now directly addressed this limitation in our discussion section. 

5) The prevalence of HCMV in these samples is a key piece of data. The data presented on page 8 (Table 1) however suggests that prevalence of HCMV is comparable to the general population and DNAemia is similar in similar sampled tissues (metastatic lung cancer, data not shown). This is a very key point to discuss since this could suggest that the findings of HCMV prevalence are no different than background population levels. This could be clarified in the Discussion where a couple of statements (i.e. third sentence of paragraph one - "suggest HCMV infection may be an unrecognized co-morbidity") are overstated.

We agree with the reviewer that the prevalence of HCMV IgG seropositivity is comparable to the general population, and that DNAemia is much higher in these cancer patients than what would be expected in the general population. We have updated the manuscript to include information from the literature on the prevalence of DNAemia in the general population, which is dramatically lower than we observe in these two patient populations. We have now added emphasis in the discussion that IgG prevalence is similar to the general population, and that active HCMV infection is much higher than expected. We have carefully reviewed the language in the discussion and revised the sentence highlighted by the reviewer to indicate that it is active HCMV infection that may be an unrecognized co-morbidity,

6) There are some minor typos/formatting/missing punctuations, including in Table 2

We thank the reviewer for this feedback and have edited the manuscript accordingly.

Reviewer #2

Major:

1. Clarity to the methods is warranted.

a. Study population: how long between the time of consent and blood sampling?

This information was not collected as part of the original epidemiology study; this limitation is now addressed in the discussion section. 

b. Serum biomarkers: Stating “…serology was performed on the Roche e411 Analyzer…”, with no additional information as to how the samples were prepared, etc. is definitely not enough information. Similarly, how were the tissue samples prepared? In the same paragraph, what does “gBa region” refer to? Is it not just “gB”? Was an uninfected control used to determine baseline for the CMV DNA qPCR? This would allow one to determine if the primers detect background (in other words, it allows one to determine the baseline). This could also be why there was no association between IgG and DNAemia (which the authors describe as ‘unexpected’ in their Discussion, p13).

We have added the requested information to the methods section. Controls, both positive and negative, were used to ensure accuracy of the assay.

c. HCMV DNA in tissue: Were the normal tissue samples also freshly frozen? What were the methods used to extract the DNA from tissue?

Yes, the normal tissue samples were also freshly frozen. DNA was extracted using a Qiagen kit. These details are now included in the methods section.

d. Lung Cancer BioBank samples: “Blood was collected at specified time points during a patient’s treatment course” - Did all patients receive the same treatment course, and did treatment regimens differ between the two study sites? What were these specified time points?

We have clarified the methods to indicate that we tested a blood sample taken at the time of study enrollment, prior to treatment. There was only one site for the enrollment of participants in the Lung Cancer BioBank.

2. Figure 1 is hard to evaluate. The lines are not labeled, making it difficult to interpret.

Figure 1 has been updated; the lines are labeled.

3. p11 – “HCMV…is highly prevalent in the global population and nearly ubiquitous in immunocompromised populations” – I would argue this is misleading, especially since the authors use the term ‘immunocompromised’ to describe transplant patients as well (who are technically immunosuppressed). Likewise, “HCMV infection may be a biomarker that identifies patients who are the most immune-compromised…” is misleading.

Thank you for these suggestions, we have edited the manuscript accordingly.

4. p12 – “…those with the longest survival had low level DNAemia” – Isn’t this not completely true when adjusted for age? Similarly, p14 (Conclusion section) – “…MPM patients with a negative or low level HCMV infection were greater than those with a high level HCM infection” – isn’t this not the case when adjusted for age? (Also, minor, but this last sentence should read “…HCMV infection”).

We have revised the text as suggested.

Minor:

1. For ease in reviewing, it would help a ton to have line numbers.

Line numbers have been added.

2. Various points in the manuscript should be clarified:

a. What is meant by “HCMV…has been observed as viremia in various cancer patients and the tumor environment in different cancers” (p4)?

We have edited the text to read: “The role of HCMV in cancer has primarily focused on the presence of virus in tumors (21,22,25–27). Less well described is the epidemiology of active HCMV infection in solid tumor cancer patients.”

b. p4 – “>50% of the population infected by ages 75-80.” This should be double-checked. It’s more like >50% by 40 years of age.

We have revised the manuscript to say “... is a highly prevalent infection with approximately 50% of the population testing seropositive by age 50”, and reference NHANES.

c. What is meant by “Less is known about the experience of HCMV viremia in cancer patients…” (p4-5)?

We have clarified and expanded this text. Our intent is to convey that active HCMV infection (viremia) has to date been understudied in cancer patients, outside the context of bone marrow transplant. Primarily the literature regarding HCMV in non-hematologic malignancy has been focused on detecting HCMV in the tumor environment.

d. The data re: 21 normal pleura samples – this is not included in the tables is it?

The reviewer is correct, the normal pleura data is only present in the text of the manuscript.

e. p10, p11, p12 – “HCMV biomarkers” should actually be defined.

This comment has been addressed.

f. p12 – what is meant by a “robust NK cell phenotype”?

This text has been clarified

3. Figures should be presented in order of appearance. As is, the supplemental figures need to be reversed. Also, I would suggest the authors put the supplemental figures in the main body of the manuscript as opposed to including them as supplemental figures.

We have corrected the order of appearance of the supplemental figures. We defer to the editor whether these should be supplemental or included in the main body of the manuscript.

4. The authors should include an overall “take-home” message for their findings at the end of the results section. As is, this section just drops off, leaving the reader wondering what the overall take is.

A brief summary of the results is now included at the end of the results section.

5. p11 – The authors state that HCMV infection could put MPM patients at risk for HCMV-associated pneumonia. Do any of these patients the authors evaluated present with this symptom?

Unfortunately, we do not have that information available. 

6. p12 – I appreciate the authors’ idea that pleural tissue may serve as a site of latency. But it may also be worth noting that infiltrating, latently infected monocytes that are recruited to the ‘injured’ pleura could be a source of reactivating virus in this tissue. Just a thought. 

We really like this idea and thank the reviewer for the suggestion.

---

## [Editor Report · Decision Letter 1]

21 Jun 2021

Cytomegalovirus infection in malignant pleural mesothelioma

PONE-D-20-34162R1

Dear Dr. Hunter-Schlichting,

We’re pleased to inform you that your manuscript has been judged scientifically suitable for publication and will be formally accepted for publication once it meets all outstanding technical requirements.

Kind regards,

Juliet V Spencer, Ph.D.

Academic Editor

PLOS ONE
---

## [Editor Report · Acceptance letter]

29 Jul 2021

PONE-D-20-34162R1 

Cytomegalovirus infection in malignant pleural mesothelioma 

Dear Dr. Hunter-Schlichting:

I'm pleased to inform you that your manuscript has been deemed suitable for publication in PLOS ONE. Congratulations! Your manuscript is now with our production department. 

Kind regards, 

on behalf of

Dr. Juliet V Spencer 

Academic Editor

PLOS ONE